# The Intersection of Intimate Partner Violence, Life Stressors, and Perinatal Loss Among Black Women from the United States: Implications for Enhancing Maternity Care Quality and Public Health Practice

**DOI:** 10.3390/ijerph22111613

**Published:** 2025-10-23

**Authors:** Jeri M. Antilla, Amy C. Buckenmeyer, Linda M. DiClemente, Madeline Carlin

**Affiliations:** 1School of Nursing, University of Michigan, Ann Arbor, MI 48109, USA; buckena@umich.edu (A.C.B.); lmdiclem@umich.edu (L.M.D.); 2College of Literature, Science, and the Arts, University of Michigan, Ann Arbor, MI 48109, USA; mpcarlin@umich.edu

**Keywords:** intimate personal violence, perinatal loss, maternal health, pregnancy, population health, perinatal outcomes, maternity care quality

## Abstract

Intimate partner violence (IPV) and life stressors, such as housing instability, unsafe neighborhoods, and lack of support, significantly impact maternal and fetal health, potentially leading to perinatal loss. This qualitative study explored the lived experiences of 22 Black women in the United States who identified IPV and other stressors as contributing factors to their perinatal loss. Semi-structured interviews were carried out with women who had experienced perinatal loss and were either pregnant or had given birth after a loss. Descriptive coding and thematic analysis were used in analyzing the data, revealing three main themes: pregnancy in the context of IPV, unsafe and unstable living environments, and challenges in finding support. Women perceived IPV and life stressors as direct causes of their loss, complicating their ability to heal and increasing their anxiety about future pregnancies. This study underscores the importance of addressing IPV and related stressors within maternity care. Maternity care providers should recognize signs of IPV and significant life stressors, provide trauma-informed, culturally responsive care, and facilitate access to supportive services. These insights inform perinatal public health strategies, including surveillance, prevention, and responsive policy.

## 1. Introduction

Intimate partner violence (IPV) is a major public health and human rights concern that affects women globally, with serious consequences for health during pregnancy and throughout life. IPV is defined as behaviors by a current or former intimate partner that inflict physical, sexual, or emotional harm, encompassing physical violence, sexual pressure, mental abuse, and controlling actions [1]. At the population level, these patterns have direct implications for perinatal public health, including surveillance, prevention, and resource allocation [1]. In 2024, the World Health Organization (WHO) reported that nearly one in three women worldwide between the ages of 15 and 49 had experienced physical or sexual violence during their lifetime. Rates of IPV are estimated to be higher in low- and middle-income countries compared to high-income countries [2]. Despite this staggering estimate, the actual magnitude of IPV is difficult to capture, as many women do not disclose their experiences due to fear, stigma, shame, cultural norms, or lack of awareness [3,4].

Experiencing IPV during pregnancy can result in adverse health outcomes for both the mother and her fetus [5]. IPV in pregnancy differs by country and cultural context, with an average prevalence rate of approximately 25% globally [6]. IPV may take several forms during pregnancy, such as physical and sexual violence, psychological or emotional abuse, financial control, and coercive behaviors. Its impact may include maternal health problems, injuries, complications, and increased risk of mental health challenges like anxiety, depression, behavioral changes, food insecurity, and poverty [3,7].

Social drivers of health (SDoH) create racial and ethnic disparities and influence IPV rates in pregnancy and birth outcomes [8,9]. Black women in the United States (U.S.) experience pregnancy-related IPV at higher rates than their White counterparts. National data indicate that approximately 10% of Black women, compared to about 5% of White women, experience IPV during pregnancy, with lifetime prevalence at nearly 45% and 25%, respectively [10,11]. IPV amplifies persistent stressors such as housing instability, unsafe neighborhoods, and lack of support, intensifying adverse health outcomes for Black women, perpetuating persistent disparities in maternal and infant health [12,13,14].

Perinatal loss refers to the unexpected or involuntary end of pregnancy, including miscarriage before 20 weeks, stillbirth after 20 weeks, or the death of a newborn within 28 days of life [15]. In the U.S., about 26% of pregnancies end in miscarriage, and up to 10% of clinically recognized pregnancies result in loss [16]. One in ten infants is born preterm, before 37 weeks of gestation [17]. Black women experience higher rates of adverse outcomes, with a preterm birth rate of 14.6% compared to 9.4% among White women [18]. These disparities extend to fetal and infant mortality, low birthweight (LBW, less than 2500 g), and very low birthweight (VLBW, less than 1500 g). In 2021, the fetal mortality rate for Black women was 9.9 per 1000 live births versus 4.8 for White women, and the infant mortality rate for Black infants remains more than twice that of White infants [19,20].

Social drivers of health, persistent stressors, and IPV contribute to these disparities [21,22,23,24]. Persistent stress from IPV, unstable housing, and unsafe neighborhoods increases allostatic load, disrupting physiologic regulation and leading to poor pregnancy outcomes [25,26,27]. Allostasis explains how the body adapts to stress by altering physiological processes, showing the effects of repeated stress on homeostasis regulation [27]. This biologic framework aligns with the “weathering” hypothesis, which extends the concept to social contexts to propose that cumulative disadvantage accelerates health deterioration, increasing the risk for hypertension, LBW, VLBW, and preterm birth [28]. Persistent stress, including IPV-related trauma, raises levels of cortisol and norepinephrine, which may cause premature labor and hinder fetal development [29,30].

IPV during pregnancy increases risk for adverse outcomes, including preterm birth, perinatal loss, low birth weight, and very low birth weight, through pathways such as injury, stress responses, and barriers to care [31]. These risks intensify when IPV co-occurs with other stressors, including housing instability, unsafe neighborhoods, and limited social support, which further constrain access to prenatal care, nutritious food, and safe environments [32,33,34]. For Black women, the burden of IPV combined with structural and social stressors creates conditions where adverse pregnancy and birth outcomes are more likely [35]. However, there remains a significant gap in research examining how persistent stressors and IPV contribute to psychological distress, shape perinatal loss experiences among Black women, and complicate subsequent pregnancies after loss. The study aim was to explore the lived experiences of Black women in the U.S. who identify IPV as a contributing factor to their perinatal loss. It seeks to understand how additional stressors, such as housing instability, unsafe neighborhoods, and lack of support, amplify the psychological and physical impacts when combined with IPV. This analysis draws on a larger qualitative dataset. This manuscript reports the subset of themes that directly address the study aim, while findings related to additional themes are presented elsewhere [12]. This analysis identifies cross-cutting themes within a cohort of Black women. We do not claim these themes are unique to Black women, and we interpret them in relation to contexts that disproportionately affect Black women in the U.S.

## 2. Materials and Methods

This study involved 22 Black women from the U.S. who had experienced perinatal loss and were either currently pregnant or had given birth following their loss. Given that 80% of miscarriages occur in the first trimester [36], we focused on losses from the second trimester (beginning at 14 weeks of gestation) up to the first 28 days after birth. This focus ensures the inclusion of clinically diagnosed pregnancies [16].

To participate, individuals needed to meet the following criteria: (1) identify as a Black woman; (2) be 18 years or older; (3) speak and read English; (4) have experienced a loss of a fetus or newborn between 14 weeks gestation and 28 days old due to miscarriage, stillbirth, or neonatal death; (5) be currently pregnant or have given birth after their loss; and (6) have experienced stress-related situations during the time of perinatal loss or in the periods leading up to and during a subsequent pregnancy.

### 2.1. Setting and Design

This study utilized Black feminist thought and a life-course perspective to understand how IPV and life stressors, such as housing instability, unsafe neighborhoods, and lack of support, uniquely impact Black women, contributing to perinatal loss. Black feminist thought emphasizes the distinct social positioning of Black women due to intersecting power structures, which can amplify their exposure to IPV and life stressors, thereby affecting their health outcomes [37]. This theoretical lens enhances understanding how complex social and environmental factors interact over the life course to influence maternal and fetal health [38]. Through a life-course perspective, researchers can explore how the places in which individuals are born, grow up, work, and age influence their health outcomes [39]. Lu and Halfon [29] suggest that health disparities, including those in pregnancy and birth outcomes, are shaped by early life experiences and the accumulated burden of stressors. When combined, these frameworks offer a comprehensive approach to exploring the intricate and situated realities faced by Black women in dealing with IPV and persistent stressors, calling for systemic changes in maternity care practices to address these issues effectively. The study aim was to explore the lived experiences of Black women in the U.S. who identify IPV as a contributing factor to their perinatal loss. It seeks to understand how additional stressors, such as housing instability, unsafe neighborhoods, and lack of support, amplify the psychological and physical impacts when combined with IPV.

For this study, we employed a descriptive and interpretive phenomenological design. This qualitative approach explores how individuals make sense of their experiences and circumstances [40]. Using inductive data analysis, the research team derived descriptive findings. This approach enabled the researchers to draw meaningful insights from the women’s narratives, organize them into recognizable patterns, and present the findings as themes that correspond with the study’s objectives and context [41].

### 2.2. Sample

Prior to the start of the study, approval was received from the primary investigator’s Institutional Review Board (University of Wisconsin-Milwaukee, IRB ID #19.A.276). Women were recruited using snowball sampling from various locations, including clinics providing women’s care, perinatal loss support groups, community centers, churches, hair salons, and social media across the U.S. Recruitment involved posting and distributing flyers at these sites. Potential participants were asked to express their interest by emailing the primary investigator, who then assessed their eligibility for the study. Those deemed eligible were instructed to complete an electronic informed consent form prior to their interview. The primary investigator was responsible for scheduling the interviews. Recruitment continued until meaning saturation was reached, at which point additional interviews did not add nuance to existing codes or alter theme definitions.

Before conducting each interview, the primary investigator provided the women with detailed information about the study’s procedures, potential risks, and their right to withdraw at any time. Information regarding the audio recording of the interview was shared. Prior to starting the interview, women were asked demographic questions to collect data on their age, geographic location, education level, annual income, employment status, relationship status, housing situation, and healthcare coverage.

The primary investigator conducted the interviews in person or via communication platforms such as FaceTime, Skype, or telephone, depending on each woman’s preference. Each interview took place at a previously arranged time using the participant’s chosen medium. At the beginning of the interview, researchers assigned numbers to each participant. This approach allowed for the correlation of demographic information while ensuring the data remained de-identified, maintaining participant confidentiality. The interviews were audio-recorded and had an average duration of 50 min. As a token of appreciation, each woman received a $25 gift card after the interview.

The study sample comprised 22 Black women aged 18 to 48 (M = 34.2, SD = 8.3), see Table 1. The women were distributed across various regions of the U.S., with the highest proportion (*n* = 10) residing in the Midwest. Other women were from the Northeast (*n* = 3), Central (*n* = 1), West (*n* = 2), and South (*n* = 6) regions of the U.S.

The women reported a median annual income of $41,500, though this amount varied widely. Seven women earned less than $20,000 yearly, while six exceeded $80,000. Employment statuses also differed; sixteen women were employed, five were unemployed, and one was a full-time student. Regarding education, one participant did not complete high school, four were high school graduates, eight had some college education, and nine held college degrees.

Relationship status varied among the women. Nine women were married, seven were single, four were divorced, one was in a relationship, and one was engaged. We did not systematically collect sexual orientation, partner gender identity, fertility treatment history, or infertility duration because these variables were outside the analytic focus of this study. Housing situations were diverse, with nine owning their homes and thirteen renting; six women were unhoused at some point during their pregnancy leading up to their loss.

Regarding healthcare coverage, 12 women utilized a federal-state program, while 10 had private insurance. The number of perinatal loss experiences among women ranged from one to eight (M = 2.1, SD = 1.6). Nine women experienced one loss, four women experienced two losses, another four experienced three losses, one experienced four losses, and one experienced eight losses. Additionally, four women endured the death of their newborn within the first 28 days after birth.

We used semi-structured, one-on-one interviews to collect data for this study. We asked the women open-ended questions and followed up with additional inquiries based on the responses. This approach allowed us to deviate from the established interview guide when necessary to better understand the women’s narratives [42,43]. The interview guide was developed from relevant literature to guide data collection. This approach enabled the gathering of comprehensive data and provided a deeper understanding of the stress factors affecting Black women who have experienced perinatal loss and subsequent pregnancies. The interview guide questions are displayed in Table 2.

Moreover, the primary investigator consistently recorded field notes before, during, and after each interview, maintaining this practice throughout the research process. These notes were essential in providing insights and helping to better understand how the women perceived and experienced stress during their perinatal loss and throughout subsequent pregnancies. Field notes are pivotal in qualitative research as they enhance the collected data and offer detailed context for further analysis [44].

### 2.3. Analysis

A professional transcription service was secured, and all audio recordings were transcribed verbatim. After transcribing was completed, we reviewed the transcriptions against the original audio files to ensure accuracy and made any necessary corrections. For confidentiality, all transcripts were de-identified and managed in NVivo 12 (QRS International) for data organization, and all qualitative coding was conducted by the primary investigator.

The analysis process involved both descriptive coding and thematic analysis. Descriptive coding summarizes data segments using words or brief phrases that capture the main topic [45]. We then conducted thematic analysis to identify overarching themes that emerged from the data [46]. This process was applied to each transcript, enabling us to identify themes and subthemes. This article will focus on data related to key themes: pregnancy in the context of IPV, unsafe and unstable living environments, and challenges in finding support. This method of analysis allowed for an in-depth exploration of women’s experiences. We used reflexive thematic analysis to identify cross-cutting patterns across the dataset and did not conduct subgroup comparisons by loss type or fertility treatment history. To validate our findings, we regularly convened via Zoom to review each code and evaluate its relevance. We carefully examined each theme and subtheme, which led to the development of comprehensive overarching themes.

We analyzed data from a larger qualitative dataset that identified multiple thematic areas, including mental wellbeing, coping, and patient-provider dynamics. For this manuscript, we report a focused analysis of three themes: pregnancy in the context of IPV, unsafe and unstable living environments, and challenges in finding support. We present findings related to other themes elsewhere [12]. Portions of the Methods section overlap with that publication because both studies use the same data-collection framework, while this analysis addresses a distinct research question and reports new thematic findings specific to the intersection of IPV, structural stressors, and perinatal loss.

#### Rigor

We employed several strategies to ensure trustworthiness. To enhance confirmability, we maintained a reflexive journal throughout data collection and analysis. Entries documented the primary investigator’s characteristics as a middle-class White woman and qualitative nurse researcher, her 26 years of obstetric nursing experience, and her outsider status to Black women’s lived experiences of perinatal loss. We used the journal to promote transparency and reduce researcher bias in data collection by noting how perceived authority, racial difference, and motherhood status might influence rapport, disclosure, probing, and other procedures. We recorded corrective steps such as rephrasing prompts, extending wait time, or scheduling a second contact. We regularly reviewed entries in team debriefs, refined interview guides, and checked that interpretations aligned with the dataset and reflected cultural safety.

After completing data collection, we solidified credibility by implementing member checking, asking women to verify the authenticity and accuracy of their transcripts. We sent each transcript to the women via secure email, resulting in 12 women confirming accuracy, while 10 did not respond. Throughout the data analysis process, we regularly utilized the expertise of our team to verify the authenticity of identified themes and subthemes. Our team met weekly or biweekly via Zoom to assess and discuss the data. To establish triangulation, we used field notes to confirm the authenticity of findings. We ensured transferability by utilizing detailed demographic data, which helped contextualize women’s stories and describe the research setting, the researchers’ roles, and research assumptions. Lastly, we enhanced dependability by conducting an audit trail to document the research process.

## 3. Results

In this article, we discuss the persistent psychological stress experienced by Black women and emphasize how factors such as IPV, housing instability, and unsafe neighborhoods can exacerbate both mental and physical health outcomes. These findings aligned with the major themes identified in this study: pregnancy in the context of IPV, unsafe and unstable living environments, and challenges in finding support. A detailed summary of each theme, corresponding subthemes, comprehensive explanations, frequency of occurrence, and relevant participant quotations for reference, is presented in Table 3. This study report’s themes that are common to women who experience IPV and perinatal loss. Our contribution is to show how women in this study situated these themes within contexts that disproportionately affect Black women.

### 3.1. Pregnancy in the Context of IPV

Women described experiencing multiple stressors in their lives before, during, and after perinatal loss, as well as during subsequent pregnancies after loss. IPV, along with concurrent apprehension and perceived stigma from maternity care providers, was reported to intensify these challenges. These stressors were captured in three subthemes: Living in Fear, Emotional and Psychological Toll, and Fear of Judgment. Many women indicated that one or both stressors were present at the time of their loss and continued into subsequent pregnancies.

#### 3.1.1. Living in Fear

Women described how IPV complicated their loss experiences and made it difficult to focus on emotional healing. For some, the abuse created a persistent sense of fear and instability that intensified anxiety and worry about the possibility of another loss. The presence of violence during and after pregnancy disrupted their feelings of safety and compounded the emotional toll of their grief. Several women believed that the abuse directly contributed to their perinatal loss, deepening both their sorrow and their ongoing psychological distress. One woman described her experience in a volatile relationship with her ex-husband, explaining:


*He choked me when I was about four months pregnant. I didn’t go to the hospital, but after that, I kept having cramps until I lost the baby.*


Another woman shared that the persistent threat of violence from her partner caused profound emotional distress throughout the period before, during, and after her perinatal loss. Several women reported that their partners’ violent behaviors began or intensified during pregnancy. Some women reported experiencing similar episodes of physical violence from their partners, while others described abuse that occurred through financial control. Several women explained that their partners restricted their financial autonomy and prevented them from seeking employment. One woman recounted feeling stressed when her partner took her money or required her to justify her spending, stating the following:


*He controlled all the money and wouldn’t let me have my own bank account. Sometimes he’d make me show receipts for everything I bought.*


For these women, this form of abuse continued both in their daily lives and during pregnancies after their loss.

#### 3.1.2. Emotional and Psychological Toll

Women described how IPV compounded their experiences of perinatal loss, creating an enduring sense of fear and emotional instability. For many, the abuse triggered anxiety, depression, and posttraumatic stress symptoms that persisted long after the loss. One woman recounted:


*The yelling and the hitting gave me PTSD. Loud noises or someone raising their voice, it takes me right back to those moments.*


Other women shared that they lived in constant anticipation of violence, feeling unsafe even in their own homes. Some described anxiety that persisted after leaving the abusive relationship, while others recounted depression so severe it made daily functioning difficult. Women spoke of the physical and emotional strain of enduring conflict during pregnancy, of feeling that their bodies could never fully relax, and of being consumed by grief and fear in the aftermath of both the abuse and the loss.

#### 3.1.3. Fear of Judgement

One woman explained that she did not attend prenatal appointments consistently because of her abusive circumstances, which added significantly to her overall stress and feelings of isolation. Several women described similar experiences, noting that the unpredictability of their partners’ behavior often made it difficult to leave the house or keep medical appointments. Some women reported avoiding healthcare visits when they had visible injuries, such as bruises or swelling, due to fear of being judged or questioned by providers. These missed or delayed appointments not only limited their access to necessary prenatal care but also intensified their anxiety about the health of their pregnancies. As one woman recalled,


*I didn’t want the doctor to see the bruises and start asking questions, so I just stayed home instead.*


### 3.2. Unstable and Unsafe Living Environments

Women described living in unsafe and unstable conditions before, during, and after perinatal loss, as well as during pregnancies that followed a loss. Homelessness and living in unsafe neighborhoods were common and identified as significant contributors to ongoing psychological strain. These environmental challenges were reflected in two subthemes: Nowhere to Call Home and Surrounded by Danger. Many women reported experiencing one or both conditions at the time of their loss, with the effects often persisting into subsequent pregnancies.

#### 3.2.1. Nowhere to Call Home

Women described housing instability as a profound stressor that shaped their perinatal loss experiences. Several women shared that they were unhoused at some point during their pregnancies, often facing unsafe or unpredictable living situations. Some women reported being displaced by family members after disclosing their pregnancies, leaving them without stable housing and with few social supports to rely on. For others, housing instability began in adolescence, such as one participant who recalled the emotional strain of being pregnant and homeless during her junior year of high school. Across these experiences, women described the dual burden of navigating pregnancy while confronting the uncertainty of where they would sleep each night. Many reported that resources were scarce, and even when support services existed, they were often difficult to access because of long waitlists, restrictive eligibility requirements, or lack of transportation. One woman described the following:


*When I got pregnant, I was already living in my car. I tried to get into a shelter, but the waitlist was months long.*


#### 3.2.2. Surrounded by Danger

Women also described unsafe neighborhoods as a persistent stressor that shaped their perinatal experiences and outcomes. Several women reported living in low-income, high-crime areas where they had grown up and continued to feel unsafe. Those who resided in neighborhoods with elevated crime rates often described being hypervigilant, such as always locking their doors or avoiding walking outside because of the threat of violence. Others lived in communities where violence was an active and visible part of daily life, contributing to a constant sense of fear and stress. For some, this violence intruded directly into their homes, further eroding their sense of security. As one woman explained,


*There were bullet holes in the walls from when people were shooting outside. I didn’t feel safe there at all.*


Neighborhood stress was intensified by limited resources within women’s communities, contributing to unsafe and unstable living environments during pregnancy. Several women described restricted access to healthcare, transportation, and nutritious, affordable foods. One woman reported that the closest hospital was more than a 30-min drive from her home, resulting in her giving birth at 17 weeks in a local urgent care facility that lacked obstetric capabilities. Others lived in areas without grocery stores offering fresh produce, relying instead on convenience stores with limited healthy options. These conditions created ongoing risk, hindered efforts to maintain a healthy pregnancy, and in some cases contributed to perinatal loss. As one woman stated,


*Everything I needed for a healthy pregnancy was far away or hard to get to, and by the time I got help, it was too late.*


### 3.3. Challenges in Finding Support

Women described feeling isolated and unsupported during experiences of IPV, housing instability, and perinatal loss. The absence of meaningful emotional, social, or practical support was identified as a significant barrier to coping and recovery. These challenges were reflected in two subthemes: No One to Turn To and Left to Cope Alone. Many women reported experiencing one or both circumstances during the time of their loss, with the effects often continuing into subsequent pregnancies.

#### 3.3.1. No One to Turn to

While many women maintained social networks that included close family and friends, they often continued to experience isolation and loneliness after their loss and during subsequent pregnancies. For some, IPV created an additional layer of stress that compounded their grief. Abusive partners often limited or prevented contact with supportive family and friends, leaving women without access to emotional or practical help. Others described choosing not to reach out because of shame, guilt, or fear related to the violence they had experienced. One woman shared the following:


*He didn’t want me talking to my family, so I just stopped calling them. After the baby died, it felt like I had nobody. I just stayed to myself.*


Several women reported having little or no social connections to rely on after their perinatal loss. For many, the combination of isolation and the ongoing effects of IPV deepened their sense of being alone in their grief. In some cases, women intentionally withdrew from others to cope with these overlapping challenges.

#### 3.3.2. Left to Cope Alone

Women described recognition of perinatal loss by family, friends, and maternity care providers as deeply important. Some women reported that their maternity care providers offered little to no follow-up after the loss, did not assess for IPV, and did not address other SDoH that shaped their circumstances. Women perceived this lack of attention to the loss and to the trauma of violence as harmful, prolonging emotional distress and carrying that burden into subsequent pregnancies. Many reported that others expected them to be strong and to manage distress on their own. When family and friends failed to acknowledge their loss and the violence they experienced, women felt their child’s life was unseen and their trauma invalidated. One woman explained the following:


*It was like nobody wanted to talk about it…not the baby, not what he did to me. It felt like they were saying none of it mattered.*


Another added:


*“I felt like I was going through everything alone, nobody was really there for me.”*


## 4. Discussion

This study used semi-structured interviews to explore how Black women who experienced perinatal loss perceive and navigate the intersection of IPV, persistent life stressors, loss, and subsequent pregnancy. Perinatal loss rarely occurs in isolation; it is compounded by IPV, housing instability, unsafe neighborhoods, and inadequate support, all shaped by structural discrimination and marginalization. Cultural stigma and institutional barriers intensify the challenges. Beyond clinical settings, these findings carry significance for perinatal public health efforts that emphasize surveillance, prevention, and resource linkage. Strengthening partnerships between maternity services and public health programs can improve continuity of care and help mitigate upstream social drivers of adverse outcomes [47,48,49,50]. These conditions underscore the urgent need for trauma-informed, culturally responsive maternity care that meets both the immediate and long-term needs of Black women facing perinatal loss within structurally inequitable systems.

### 4.1. The Unseen Toll of IPV on Pregnancy and Loss

IPV profoundly affected perinatal experiences for Black women in this study. Ongoing abuse and threats from partners generated persistent anxiety and a pervasive sense of insecurity that extended throughout pregnancy and after perinatal loss. This ongoing stress has been linked to higher risks of miscarriage, preterm birth, low birthweight, and perinatal loss, as documented in recent research [31,35]. For many women, exposure to IPV during pregnancy not only contributed to immediate adverse health outcomes but also complicated the emotional recovery needed for future pregnancies.

The continued impact of IPV often extends into subsequent pregnancies following perinatal loss, contributing to ongoing psychological distress and heightened anxiety about future reproductive outcomes. Women who experience IPV during or after pregnancy frequently report persistent emotional challenges, such as worry, trauma, and fear, which can influence both their mental health and health-related behaviors [47,48]. These unresolved effects of IPV may lead women to reduce engagement with prenatal care and feel reluctant to seek necessary support. These factors can elevate risks to both maternal and infant health [49,50]. The relationship between unresolved trauma, inadequate support, and continued exposure to IPV underscores the critical need for trauma-informed maternity care that addresses these intersecting risks across the perinatal period.

Additionally, women contended with a significant emotional toll as they coped with the aftermath of both violence and loss. The fear of negative judgment from maternity care providers, family, or the broader community often led women to conceal their experiences, hesitate to seek help, or postpone accessing needed care. This reluctance deepened their grief and further isolated them during and after their loss [51]. Continued exposure to IPV has also been associated with symptoms of depression, anxiety, and posttraumatic stress, which can persist in subsequent pregnancies, further increasing risk and emotional distress [52]. These findings highlight the importance of embedding routine IPV screening, trauma-informed care, and culturally responsive support into perinatal services to address both immediate and lasting impacts of IPV on maternal and infant health.

### 4.2. Housing Insecurity and Unsafe Spaces

Unstable housing and unsafe neighborhoods emerged as major contributors to perinatal loss and psychological distress for Black women in this study. Women described how constant transitions, lack of secure shelter, and fear in their surroundings made it challenging to engage in consistent prenatal care and maintain emotional wellbeing. Safety concerns, unreliable access to nutrition, and transportation barriers often took priority over seeking medical support, resulting in missed or delayed appointments and increased vulnerability to negative health outcomes [35,53]. The ongoing stress of housing instability intensified the emotional strain of pregnancy and complicated grief after perinatal loss.

In addition to unstable housing, many women detailed the impact of living in high-crime neighborhoods, which led to hypervigilance, anxiety, and withdrawal from community or health resources. Environmental and psychosocial stressors in these unsafe spaces are associated with elevated allostatic load, higher maternal mental health problems, and increased risk of adverse pregnancy outcomes through mechanisms such as increased cortisol levels and persistent stress [35,54,55]. Recent studies further demonstrate that maternal perceptions of neighborhood danger are linked to depression, preterm birth, and diminished psychological wellbeing [54,56,57]. These findings align with previous research showing that structural inequities related to poverty and segregation amplify the difficulties faced by Black women during pregnancy, emphasizing the need for integrated strategies that address housing security and community safety within perinatal care [32].

Unsafe environments combined with barriers to healthcare make structural factors such as poverty, racial segregation, and systemic inequities even more pronounced. These conditions further exacerbate disparities in perinatal outcomes and psychological wellbeing [32]. These interrelated challenges highlight the need to integrate housing stability and community safety interventions into perinatal care to improve outcomes for vulnerable populations.

### 4.3. Barriers to Comfort and Connection

The study findings indicate that when women lack consistent social support after perinatal loss, the resulting isolation significantly heightens their vulnerability to depression, anxiety, and complicated grief [58,59]. Factors such as partner abuse, community stigma, and broader cultural pressures can contribute to this lack of support. This compounded risk particularly affects Black women, who already face systemic barriers in accessing mental health and bereavement support. Research demonstrates that these mental health challenges can impede maternal recovery and lead to lasting morbidity and mortality [60]. For this reason, experts highlight the importance of responsive, timely mental health screening and interventions that are accessible and culturally tailored to the needs of Black women [51]. Addressing stigma and building supportive care networks are critical steps in reducing disparities and improving outcomes after perinatal loss.

Feelings of social isolation and IPV hindered women from openly discussing their loss, which further increased their anxiety and depressive symptoms and ultimately diminished their wellbeing. Cultural patterns within Black families often inhibit conversations about IPV, perinatal loss, and related emotions, prompting women to internalize fears about ongoing abuse and possible consequences for future pregnancies. For example, many families and communities expect Black women to embody the “strong Black woman” (SBW) archetype [58]. This cultural ideal emphasizes the importance of displaying strength, suppressing emotions, persevering through hardship, and prioritizing the needs of others [61,62]. Though some view SBW beliefs as affirming, others argue they oversimplify Black women’s lived realities and may limit emotional expression or help-seeking behaviors [63,64,65]. Adhering to this schema can increase stress and influence coping [66]. The SBW ideal arises from the intersectionality of racial and gender identities and sets expectations for resilience and self-sacrifice [61,62]. These expectations can conflict with the experiences of Black women navigating IPV and perinatal loss.

### 4.4. Strengths and Limitations

This study offers important strengths, including in-depth qualitative insights into the lived experiences of Black women facing perinatal loss and IPV. The study provides rich data on the interplay between IPV, stressors, and pregnancy outcomes by recruiting women from diverse backgrounds and using detailed, semi-structured interviews. Employing rigorous analytic methods and triangulating findings across different modes of data collection enhances the credibility and trustworthiness of the results. These strengths enhance the valuable knowledge gained through this study in an underexplored area, supporting the need for more inclusive, culturally responsive, trauma-informed perinatal care.

This study has several limitations. Most women lived in the Midwest and Eastern U.S., so the findings may not fully represent Black women in other regions, especially the South, where sociopolitical and resource contexts may differ. Although regional differences exist, overall stress patterns and pregnancy effects likely remain similar nationwide. Conducting thirteen interviews by telephone limited opportunities to observe nonverbal cues and in-person interactions, which may have reduced data depth. However, the themes identified were consistent across data collection methods. The relatively small sample size, while suitable for phenomenological research, may affect transferability to all Black women who have experienced perinatal loss linked to IPV and related stressors. Only 12 women reviewed their transcripts for accuracy, and although the remaining 10 did not, the consistency of themes across interviews helps mitigate concerns about rigor. Requiring English proficiency may have excluded women who do not speak or read English. As with all qualitative studies, the results reflect personal narratives and depend on women’s willingness to disclose sensitive experiences. We did not characterize women’s sexual orientation or partner gender identity, which may limit transferability to LGBTQ+ populations and relationship configurations. The sample size and analytic focus precluded comparisons by loss type or fertility treatment history, which may limit transferability to specific subgroups. We did not conduct subgroup analyses comparing Black women with other racial or ethnic groups. We do not assert that the reported themes are unique to Black women. Instead, we describe how these themes manifested within this cohort and context. Future research should purposively collect sexual orientation and partner gender identity, record fertility treatment history and infertility duration, and recruit sufficient samples across racial and ethnic groups to examine differences by loss type and pregnancy circumstances.

### 4.5. Implications and Recommendations

The findings from this study underscore how IPV, together with structural and social stressors reflected in the SDoH, shaped women’s experiences of perinatal loss and subsequent pregnancy. Key stressors included housing instability, unsafe neighborhood conditions, limited social support, and fear of judgment in clinical settings. These intersecting factors can elevate stress, adversely influence future pregnancies, and contribute to adverse maternal and neonatal outcomes [23]. The findings in this study should be read as themes observed in a cohort of Black women and interpreted in relation to racialized social and structural conditions, rather than as phenomena unique to Black women. The recommendations that follow target the needs described by the women and are intended to guide actionable change for settings that care for Black women experiencing perinatal loss in the context of IPV.

A person-centered, trauma-informed, and culturally responsive approach is critical to improving outcomes for Black women at risk for or who have experienced perinatal loss [67]. Culturally responsive care includes offering access to Black clinicians or paraprofessionals when available, providing peer-led perinatal loss groups for Black women, and coordinating transportation, childcare, and safety-planning support so women are not left to navigate services alone. Policy initiatives that promote individualized interventions, including reproductive life planning, empower women to address IPV and improve reproductive health across the life course, ultimately strengthening maternity care systems to better meet the needs of vulnerable populations [3,68,69].

Continuity of care with trusted providers can enhance outcomes. Women who fear judgment, particularly in the context of IPV-related injuries, may delay or avoid prenatal care [70,71,72]. Assigning consistent maternity care providers fosters trust, encourages open communication, and strengthens adherence to care [73,74]. Training maternity care providers in cultural safety and bias recognition is necessary to create nonjudgmental environments that support disclosure of IPV and related stressors [75,76]. This training should include scripted, trauma informed IPV inquiry with role play and feedback, case reviews that examine racialized clinical encounters, reflective supervision focused on countering stigma and victim blaming, routine warm handoffs rather than passive referrals, and location enabled resource lists so staff can connect women during the visit to nearby housing, transportation, food, and legal supports. Post-perinatal loss follow-up and bereavement care must also be prioritized. Immediate outreach through phone or telehealth, paired with trauma-informed care, can reduce fear and foster disclosure while addressing the psychological burden of IPV and loss [77,78].

Specialized bereavement support that reflects the cultural needs of Black women, as described by women in this study, includes proactive outreach within one to two weeks after loss, acknowledgement of the baby by name when desired, and repeated, nonjudgmental inquiry about safety and IPV. It also includes warm handoffs to grief and IPV resources, options for peer support groups led by Black loss counselors or doulas, flexible in-person and telehealth visits outside standard hours, and the option to include faith leaders or trusted supporters if the woman wishes. These supports validate women’s experiences, promote healing, and support emotional recovery and readiness for future pregnancies [79].

Early and ongoing screening for IPV and related stressors, including housing instability, unsafe neighborhoods, food insecurity, and transportation barriers, is essential. Given that circumstances often change during pregnancy and postpartum, screening must be repeated across the continuum of care and across public health programs [80]. Screening should include immediate referral pathways and warm handoffs to resources, ensuring that women are not left to navigate fragmented systems alone. Public health agencies can maintain geo-coded, up-to-date resource directories, and enable bidirectional e-referrals between clinics and community providers to prevent loss to follow-up.

Housing stability and safe neighborhoods are fundamental to maternal health, and partnerships between healthcare and social service agencies can expand access to safe housing, nutritious foods, and reliable transportation [81,82]. Health departments can operationalize cross-sector collaborations with housing authorities, transportation agencies, and community-based organizations to address these upstream drivers as part of perinatal health improvement plans. Equally important is mental health support, as women reported untreated anxiety and depression following perinatal loss and during subsequent pregnancies. Integrating bereavement and IPV-informed counseling within home visiting, group prenatal care, and community health worker programs extends reach beyond clinics and normalizes help-seeking. Expanding behavioral health screening and integrating culturally responsive services, including telehealth and peer support, can reduce stigma and improve access to care [83,84].

Care teams should pair system-level screening with strategies that counter women’s reported pressure to cope alone. Targeted communications from health agencies and community programs can reduce self-silencing. Paired with peer-led supports, these efforts make it easier to seek care while protecting privacy and safety. Many women in this study recounted expectations of strength and self-management of distress. Maternity and mental health providers should explicitly normalize help-seeking, invite a support person to health visits when desired, and use direct prompts that assess isolation and self-silencing. Continuity with a single trusted provider, combined with culturally safe communication and proactive bereavement follow-up, can support disclosure, reduce fear, and improve engagement in care.

Future studies should further investigate the interconnected effects of IPV, housing instability, and unsafe neighborhoods on perinatal loss among Black women. Longitudinal research is needed to clarify the impact of persistent stress, limited resources, and repeated exposure to life stressors on pregnancy outcomes. Researchers should develop and evaluate culturally tailored screening tools, protocols for continuous assessment throughout pregnancy and postpartum, and effective referral systems. To improve screening and enhance continuity of care, virtual perinatal and infant nurse health coaching represents a promising service delivery model for identifying IPV risk factors and providing tailored referrals to mental health providers and other needed services [85]. Nurse health coaches who integrate social drivers of health into care planning can intervene upstream by identifying and addressing factors such as housing instability, unsafe neighborhoods, food insecurity, and transportation barriers that increase vulnerability and stress within families [85]. By assessing clients’ social drivers through nurse health coaching, nurses can leverage artificial intelligence to geospatially connect families with nearby social, financial, and health resources [86]. In addition, targeted reproductive life planning interventions and culturally responsive bereavement and mental health support models such as telehealth, peer-led initiatives, and integrated behavioral health care deserve further study. Research should also assess how maternity care provider training in cultural safety, bias recognition, and trauma-informed care influences patient trust and healthcare engagement. Such training may also shape equity in maternal health outcomes. By evaluating the scalability and sustainability of these strategies, future work can identify best practices and help reduce perinatal health disparities linked to IPV and persistent stressors.

## 5. Conclusions

This study highlights stressors including IPV, housing instability, unsafe neighborhoods, and lack of support on the perinatal experiences of Black women in the U.S., with women perceiving these factors as directly contributing to perinatal loss and influencing subsequent pregnancies [15,22,87,88]. This analysis is relevant to both maternity services and public health, identifying clinical practice changes and population-level prevention that address shared risk pathways linking IPV, structural stressors, and perinatal loss. To address the intensifying effects of these intersecting stressors on psychological and physical health, maternity services should implement unbiased, culturally responsive, trauma-informed care with comprehensive, ongoing screening across the reproductive continuum [73,84]. Targeted interventions such as reproductive life planning [68], improved housing and community resources [71,72], and accessible mental health and bereavement support [74,84] are critical for reducing disparities in perinatal outcomes and fostering maternal and newborn wellbeing [24,28]. These findings reflect themes observed in a cohort of Black women and should be interpreted in relation to racialized social and structural conditions, not as phenomena unique to Black women. They point to actionable changes that women in this study identified as helpful, including proactive bereavement follow-up, culturally responsive supports such as access to Black clinicians or peer groups, and training that builds cultural safety and bias recognition in maternity care. Future work should test these approaches across racial and ethnic groups and examine differences by loss type and pregnancy circumstances.

## Figures and Tables

**Table 1 ijerph-22-01613-t001:** Sociodemographic and Perinatal Characteristics of Women (*n* = 22).

Age: mean 33.1, range 18 to 48 years Region: Midwest (10), South (6), Northeast (3), West (2), Central (1) Housing: renting 13, homeowner 9 Education: <HS (1), HS (5), some college (7), college degree (9) Income: <20 K (8), 20–30 K (5), 30–40 K (1), 40–50 K (2), 60–70 K (1), >80 K (5) Relationship: married (9), single (7), divorced (4), engaged (1), in a relationship (1) Employment: employed (16), unemployed (5), full-time student (1) Pregnancies: median 4, range 1 to 10 or more Perinatal losses: median 2, range 1 to 4 or mor GA at loss: <14 weeks (14), >14 weeks (9) NB at loss: <28 days (3), >28 days (2)

Note. Women’s demographics are drawn from a larger qualitative dataset [12] and are summarized and reformatted for clarity in this manuscript. Income ranges are in USD. GA, gestational age; NB, Newborn; HS, high school.

**Table 2 ijerph-22-01613-t002:** Key Interview Topics and Illustrative Prompts.

Topic Area	Illustrative Prompt
Participation Motivation	What motivated you to take part in this research?
Impact of Stress & Sources of Support	Can you describe some of the pressures or challenges you face in your daily life?
	How have the pressures or challenges in your life affected this or previous pregnancies?
	Who do you turn to when you need to talk about your stress or emotions?
Factors Shaping Pregnancy Loss Experience	Please share what you remember about the pregnancy that ended in loss. Describe the kind of prenatal care you received during that pregnancy. What kind of care or support did you receive after your loss, and how well did it meet your needs? How did different circumstances or influences shape your experience of loss? What additional stressors made your grieving process more difficult after your loss? What helped you find comfort or manage your emotions following your baby’s death?
Subsequent Pregnancy, Emotions, & Connection with Baby	Tell me about your most recent or current pregnancy experience. How did you feel when you first learned you were pregnant again? What aspects of this pregnancy have been the most difficult for you?” How would you describe the connection you felt with your baby during this pregnancy? What kinds of stress have you faced throughout your current or most recent pregnancy? Tell me about the prenatal care you received during this pregnancy?” What emotions have stood out for you during this current or recent pregnancy?
Empowerment and Reflection	What would you want other mothers who have experienced loss to know or take away from your experience?

Note. Interview prompts were adapted and rephrased from a larger qualitative dataset [12] to fit the current analytic focus.

**Table 3 ijerph-22-01613-t003:** Themes, Subthemes, Definitions, and Illustrative Quotes (*n* = 22).

Theme	Subtheme	Definition/Summary	Illustrative Quote
**1. Pregnancy in the Context of IPV** (*n* = 16)	Living in Fear	Fear and anxiety during and after pregnancy, shaped by IPV and threat of violence or loss	“When I was pregnant, he started hitting me. It got worse after the baby came.” “He didn’t hit me often, but when he did, it was bad. One time I ended up with bruises all over my arms, and not long after that, I lost the baby.” “He used to tell me I was nothing without him and no one would believe me if I said he hit me.”
	Emotional and Psychological Toll	Persistent emotional struggles, including trauma, depression, PTSD	“He told me if I ever left him, he would take the baby, and I’d never see her again.” “I had constant anxiety during that pregnancy because of the fights and the stress at home. My body never felt safe.” “Depression just took over after the abuse and the loss. I didn’t want to be around anyone, and I couldn’t sleep without nightmares.”
	Fear of Judgment	Avoiding care or disclosing abuse due to shame or fear of stigma	“I didn’t want the doctor to see the bruises and start asking questions, so I just stayed home instead.”
**2. Unsafe and Unstable Living Environments** (*n* = 18)	Nowhere to Call Home	Homelessness, unstable housing, and frequent moves during pregnancy or loss	“We were staying in my cousin’s basement with no heat. I was pregnant, cold all the time, and stressed about where we were going to live.”
	Surrounded by Danger	Neighborhood violence, lack of safe and nutritious resources	“We were living in the projects, and it was dangerous- shootings, police raids, all of that while I was pregnant.” “There wasn’t no grocery store near me, just the gas station. So, I ate whatever they had, mostly chips and pop, even when I was pregnant.” “The closest hospital was like 40 min away, so when I went into labor early, I ended up at this urgent care that couldn’t do nothing for the baby.”
**3. Challenges in Finding Support** (*n* = 9)	Nowhere to Turn	Lacking emotional, social, or practical support	“I felt like I was going through everything alone, nobody was really there for me.”
	Left to Cope Alone	Stigma, isolation, and limited access to help	“My family wasn’t around, and he wasn’t supportive, so I had to handle the pregnancy by myself.”

## Data Availability

The data from this study is not publicly available, as participants did not provide consent for sharing beyond the research team.

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
