# Peer review of "The Intersection of Intimate Partner Violence, Life Stressors, and Perinatal Loss Among Black Women from the United States: Implications for Enhancing Maternity Care Quality and Public Health Practice"

_ijerph, 2025, doi:10.3390/ijerph22111613_

Round 1

Reviewer 1 Report

Comments and Suggestions for Authors

Dear Authors,

Thank you so much for this great article, "The Intersection of Intimate Partner Violence, Life Stressors, and Perinatal Loss Among Black Women from the United States: Implications for Enhancing Maternity Care Quality." Your article provides significant contributions to the evidence around these critical topics. The paper aims to explore the counter-narratives of Black women and their experiences of IPV and perinatal loss. The paper's strengths lie in its effective centering of the voices of Black women, a group that has been historically neglected in perinatal and IPV research. I have the following recommendations for strengthening your manuscript:

  1. In the Methods section, under rigor, there was a mention of reflexive journaling to reduce researcher bias. I would recommend including more description, such as personal characteristics, and how and if this influenced the interview dynamics and the relationship with the participants and the research team. I recommend the Consolidated Criteria for Reporting Qualitative Research (COREQ) or SRQR checklist, which provides guiding questions for this piece.
  2. Under rigor or data analysis, how many coders coded the data?
  3. For sample and data collection, there were 22 participants. Did you have a study stop date, or did you meet saturation for recruitment? I would recommend a statement on this.

Other than the few comments above, the results and discussion section provided a great ending to the paper. Thank you for this work; I look forward to seeing it published.

Author Response

Reviewer 1 comment: In the Methods section, under rigor, there was a mention of reflexive journaling to reduce researcher bias. I would recommend including more description, such as personal characteristics, and how and if this influenced the interview dynamics and the relationship with the participants and the research team. I recommend the Consolidated Criteria for Reporting Qualitative Research (COREQ) or SRQR checklist, which provides guiding questions for this piece.

Author response: Thank you for this guidance. We added a positionality statement in the Rigor subsection that specifies the primary investigator’s personal characteristics and describes how these factors informed interview dynamics, rapport, and team interactions (lines 262-272). We also clarified how we used the reflexive journal to document and address these influences throughout data collection and analysis. In addition, we completed and uploaded the COREQ checklist as supplementary material and cross-checked our reporting to ensure comprehensive coverage.

Reviewer 1 comment: Under rigor or data analysis, how many coders coded the data?

Author response: Thank you for the question. All qualitative coding was conducted by the primary investigator (see Methods, lines 235–237). The research team met regularly to review code-data extracts and collaboratively develop and refine themes.

Reviewer 1 comment: For sample and data collection, there were 22 participants. Did you have a study stop date, or did you meet saturation for recruitment? I would recommend a statement on this.

Author response: Thank you for the suggestion. Recruitment proceeded iteratively and stopped when saturation was reached. No new codes emerged in later interviews, and subsequent interviews confirmed the existing codebook (see Methods, lines 154-155). We have clarified this in the manuscript.

Reviewer 2 Report

Comments and Suggestions for Authors

Thank you for your paper on IPV, Life Stressors and Perinatal Loss Among Black Women in the United States. I was looking forward to reading this as I do think it's a highly understudied area (despite how often it is acknowledged that this population faces distinct risks). However, overall, I was surprised with the direction that your paper took. The title, abstract and introduction led me to believe that how Blackness, ethnicity and/or race would be a central part of your research, yet, your results did not really include this at all. For example, your themes and indicative quotes didn't really offer anything that seemed specifically related to their identity as a Black woman. To be clear, I'm not saying that quotes or themes would have to explicitly reference race or ethnicity at every turn but, if I was provided no context for this sample, nothing included in Table 3 would be unique to your sample, rather, the experience of many women subjected to IPV during pregnancy. If this was intentional, then I would suggest making this clear and reflecting on that.

That being said, I do appreciate that Black women may disproportionately face some of the issues discussed, but it would be difficult for you to say, in the context of your study, that this was because your participants were Black (rather than another race or ethnicity).

Admittedly, I hesitate to comment on a study's design yet I was surprised that questions did not sufficiently support exploring participants' relationship to their race and ethnicity and how that might relate to some of the wider study ideas. For example, if you wanted to offer concrete suggestions on enhancing maternity care, why not ask the participants about that? I noted that you ask about what they might like to share, but why not explicitly ask this?

I have included a few further comments below. 

"Research on IPV during pregnancy has documented its association with physical harm and psychological stress": I found this statement a little confusing. How could IPV not be associated with physical harm and psychological stress? Perhaps rephrase so its less redundant.

Please excuse my ignorance but is it not the case that, statistically speaking, Black women in America are already disproportionately facing some of the stressors listed (e.g., housing instability, unsafe neighbourhoods etc.), would it be important to acknowledge this? I see this is noted in terms of perinatal loss and adverse outcomes and it is mentioned that SDoH create said disparities but it might be helpful to reiterate this when discussing the "other stressors".

Was there any concern that there might be nuances between women who experienced a loss that was miscarriage, stillbirth or neonatal death, firstly, but, equally, secondly, the relationship between that loss and IPV and, perhaps thirdly, the circumstances surrounding the pregnancy (e.g., IVF, several periods of infertility etc.)?

Did you consider asking the women their sexual orientation and/or whether their partner was a man, women or non-binary?

"The findings of this study underscore how IPV, compounded by life stressors such as race-based disparities, shapes the perinatal loss experiences of Black women.": based on what you've included, in terms of data, I'm not sure you can say this. Again, I want to be clear, I wholeheartedly believe that this is accurate but not necessarily reflected in the data you've presented.

For example, you talk about the "cultural needs of Black women" - though this may be true, where is the data from your participants reflecting on this. Equally, you refer to the archetype of the "strong black woman", did that feature in conversations (or something similar to that idea)? I was further surprised there was no mention of systemic failures to provide pain relief to Black women, relationship between religion and so-called "modern" medicine, potential distrust of Western medicine/White healthcare providers etc. 

Author Response

Reviewer 2 comment: Thank you for your paper on IPV, Life Stressors and Perinatal Loss Among Black Women in the United States. I was looking forward to reading this as I do think it's a highly understudied area (despite how often it is acknowledged that this population faces distinct risks). However, overall, I was surprised with the direction that your paper took. The title, abstract and introduction led me to believe that how Blackness, ethnicity and/or race would be a central part of your research, yet, your results did not really include this at all. For example, your themes and indicative quotes didn't really offer anything that seemed specifically related to their identity as a Black woman. To be clear, I'm not saying that quotes or themes would have to explicitly reference race or ethnicity at every turn but, if I was provided no context for this sample, nothing included in Table 3 would be unique to your sample, rather, the experience of many women subjected to IPV during pregnancy. If this was intentional, then I would suggest making this clear and reflecting on that.

That being said, I do appreciate that Black women may disproportionately face some of the issues discussed, but it would be difficult for you to say, in the context of your study, that this was because your participants were Black (rather than another race or ethnicity).

Admittedly, I hesitate to comment on a study's design yet I was surprised that questions did not sufficiently support exploring participants' relationship to their race and ethnicity and how that might relate to some of the wider study ideas. For example, if you wanted to offer concrete suggestions on enhancing maternity care, why not ask the participants about that? I noted that you ask about what they might like to share, but why not explicitly ask this?

Author response: Thank you for engaging closely with the intent and scope of our study. We address each point below.

  • Aim and focus of the analysis

The study aim was to explore how intimate partner violence, in combination with additional stressors such as housing instability, unsafe environments, and limited support, shaped perinatal loss experiences among Black women and influenced subsequent pregnancies. Our analysis was designed to identify cross-cutting patterns in those experiences within a sample of Black women, rather than to conduct between-group comparisons or to require participants to frame their accounts explicitly through racial identity language. The contribution of this manuscript is to center Black women’s narratives about the co-occurrence of IPV and social determinants of health and how that configuration shaped their experiences around loss and later pregnancy. We do not claim that every coded theme is unique to Black women; we report what was salient in this cohort, consistent with the stated aim.

  • Race, SDoH, and what the dataset supports

We agree that Black women in the United States disproportionately encounter structural and social stressors captured within the social determinants of health. Our manuscript situates the findings in that established evidence base in the Introduction and Implications. In the Results, we present what participants described: housing instability, unsafe neighborhoods, limited support, and avoidance of care due to anticipated judgment. These domains are recognized pathways through which racialized inequities are lived, and they were central to participants’ accounts, even when participants did not use race-explicit language. We intentionally avoided essentializing or asking participants to perform racial explanation on demand; instead, we allowed identity to surface where participants chose to foreground it and focused analytically on the mechanisms they described.

  • On the expectation for identity-explicit quotes

We respect the concern that quotes did not always name race explicitly. In reflexive thematic analysis, identity can operate implicitly through context, constraint, and access to resources as described by participants. Our Results reflect those mechanisms rather than imposing a requirement that participants narrate racial identity in every segment. We believe this approach is consistent with our aim and with trauma-informed interviewing that avoids leading questions or placing additional explanatory burden on participants.

  • Interview guide and practice implications

The interview guide elicited experiences of loss in the context of IPV and probed barriers, supports, and interactions with services. Participants volunteered needs, barriers, and helpful practices, and those accounts informed the practice and policy implications we present. We synthesized these participant-driven insights into recommendations rather than administering a prescriptive checklist of “what should providers do,” in order to reduce demand characteristics and allow unanticipated domains to emerge. We view this as a strength of the qualitative design.

  • Scope of reported themes

As noted in the Introduction and Analysis, the current manuscript reports three focal themes aligned with the aim. Additional thematic areas from the larger dataset, including provider-specific dynamics, are presented in a companion manuscript. This scoped reporting explains why some topics are not elaborated here, despite their relevance to the broader phenomenon.

Reviewer 2 comment: "Research on IPV during pregnancy has documented its association with physical harm and psychological stress": I found this statement a little confusing. How could IPV not be associated with physical harm and psychological stress? Perhaps rephrase so its less redundant.

Author response: Thank you for noting the redundancy. We revised the text to state the outcomes directly and remove the obvious intermediary phrasing. See lines 88-93

Reviewer 2 comment: Please excuse my ignorance but is it not the case that, statistically speaking, Black women in America are already disproportionately facing some of the stressors listed (e.g., housing instability, unsafe neighbourhoods etc.), would it be important to acknowledge this? I see this is noted in terms of perinatal loss and adverse outcomes and it is mentioned that SDoH create said disparities but it might be helpful to reiterate this when discussing the "other stressors".

We agree that it is important to make the provenance and scope of these stressors explicit. Our analysis for this manuscript focuses on three themes aligned with the study aim. Broader stressor domains and provider-focused dynamics from the same study are reported in a companion manuscript. To make this clear to readers, we added language in both the Introduction and the Analysis sections indicating that the present report draws on a larger dataset and that additional themes are presented elsewhere. See lines 102-104 and 250-254.

Reviewer 2 comment: Was there any concern that there might be nuances between women who experienced a loss that was miscarriage, stillbirth or neonatal death, firstly, but, equally, secondly, the relationship between that loss and IPV and, perhaps thirdly, the circumstances surrounding the pregnancy (e.g., IVF, several periods of infertility etc.)?

Author response: Thank you for raising the question of heterogeneity. Our design and analytic approach addressed this in three ways:

  • Variation in loss type.

The sample included a range of losses within our inclusion window (second-trimester loss beginning at 14 weeks through 28 days postpartum). We analyzed narratives using reflexive thematic analysis to identify cross-cutting patterns across the corpus rather than to compare subgroups by loss type. Given the sample size, the study was not powered for between-group analyses, and we therefore did not make claims that hinge on differences between miscarriage, stillbirth, and neonatal death. We clarified this scope in the Methods (Analytic Approach) and note it as a limitation. See lines 245-247.

  • Relationship between IPV and loss.

Consistent with the study aim, participants self-identified IPV as part of their perinatal loss experience. We examined how IPV co-occurred with stressors such as housing instability, unsafe environments, limited support, and fear of judgment in clinical settings, and how these factors shaped experiences of loss and subsequent pregnancy. We did not attempt causal attribution of IPV to specific obstetric outcomes by loss type. We clarified this sentence in the Methods to state that causal comparisons by loss category were outside the study’s scope. See line 181-184.

  • Pregnancy circumstances (for example, IVF or infertility).

We did not systematically elicit or record fertility treatment history or infertility duration, because these were not focal analytic categories for the present aim. When participants volunteered such information, it was used as context within their individual narratives rather than as a basis for subgroup analysis. We added this detail to Methods (Participants/Data Collection) and acknowledge in Limitations that unmeasured pregnancy circumstances may introduce nuance the current analysis does not disaggregate. See lines 556-562.

Reviewer 2 comment: Did you consider asking the women their sexual orientation and/or whether their partner was a man, women or non-binary?

Author response: We appreciate this suggestion. The study’s aim was to explore the lived experiences of Black women in the U.S. who identify IPV as a contributing factor to their perinatal loss. It seeks to understand how additional stressors, such as housing instability, unsafe neighborhoods, and lack of support, amplify the psychological and physical impacts when combined with IPV. Sexual orientation and partner gender identity were not focal analytic categories in this design, and we did not systematically collect these variables. Some participants referred to partners using gendered language in their narratives, but because this information was not elicited consistently, we did not analyze or report partner gender or sexual orientation. 

To address this point transparently, we added the following clarifications:

Methods, Participants/Data Collection: “We did not collect sexual orientation or partner gender identity systematically because these were not focal analytic categories for this study’s aim.” See lines 179-182

Limitations: “We did not characterize women’s sexual orientation or partner gender identity, which may limit transferability to LGBTQ+ populations and relationship configurations. Future research should purposively include and analyze sexual orientation and partner gender identity to deepen understanding of IPV, loss, and care needs across diverse families.” See lines 556-562.

Reviewer 2 comment: "The findings of this study underscore how IPV, compounded by life stressors such as race-based disparities, shapes the perinatal loss experiences of Black women.": based on what you've included, in terms of data, I'm not sure you can say this. Again, I want to be clear, I wholeheartedly believe that this is accurate but not necessarily reflected in the data you've presented.

Author response: We appreciate the careful reading. The statement aligns directly with the study aim, which was to explore how intimate partner violence (IPV) and additional stressors such as housing instability, unsafe neighborhoods, and lack of support amplify psychological and physical impacts for Black women following perinatal loss. The data we present include participants’ accounts of housing instability, unsafe environments, limited support, and avoidance of care due to anticipated judgment. These are social drivers of health (SDoH) domains that operate as mechanisms through which racialized inequities are experienced, without requiring us to assign stereotypes or narrate structural histories in this manuscript.

To avoid implying that we directly measured structural racism as a construct in this paper, we revised the sentence, so it explicitly references the SDoH domains evidenced in participants’ narratives while retaining the focus of the aim. See lines 565-569.

Reviewer 2 comment: For example, you talk about the "cultural needs of Black women" - though this may be true, where is the data from your participants reflecting on this. Equally, you refer to the archetype of the "strong black woman", did that feature in conversations (or something similar to that idea)? I was further surprised there was no mention of systemic failures to provide pain relief to Black women, relationship between religion and so-called "modern" medicine, potential distrust of Western medicine/White healthcare providers etc. 

Author response: We appreciate the emphasis on grounding interpretive claims in participants’ accounts. The present manuscript already includes participant testimony speaking to cultural safety needs, expectations to “be strong” or cope alone, and care avoidance driven by fear of judgment. We also note that provider-specific dynamics beyond the three focal themes are addressed in a companion analysis from the larger dataset (Antilla & Johnson, 2024).

Culturally responsive care needs are participant-driven in this manuscript.

In Implications we recommend culturally responsive, trauma-informed maternity and bereavement care tailored to Black women. Those practice points explicitly arise from participants’ descriptions of stigmatizing or culturally unsafe encounters and their stated needs for acknowledgement and follow-up after loss. For example, women described lack of recognition and follow-up from maternity services as harmful, which is why we point to culturally safe bereavement support “that reflects the cultural needs of Black women.”

Expectations to be strong and to cope alone are documented in participants’ words.

While participants did not always name the “Strong Black Woman” label explicitly, they repeatedly described pressures and contexts that map onto that schema: isolation, withholding distress, and handling everything without support. Illustrative participant quotations include “I felt like I was going through everything alone, nobody was really there for me,” and “He didn’t want me talking to my family, so I just stopped calling them… it felt like I had nobody.” These appear in the “Challenges in Finding Support” theme.

Mistrust and care avoidance are already presented.

Participants reported avoiding prenatal visits when they feared judgment about visible injuries: “I didn’t want the doctor to see the bruises and start asking questions, so I just stayed home instead.” We also note apprehension and perceived stigma from maternity providers within the IPV theme. Together these findings speak directly to trust and disclosure in clinical settings.

Scope of this paper vs the larger dataset.

As stated in the Introduction and Analysis, this manuscript reports three focal themes that align with the study aim: pregnancy in the context of IPV, unsafe and unstable living environments, and challenges in finding support. Other thematic areas from the larger dataset, including patient–provider dynamics, are reported elsewhere, which is why detailed treatment of topics such as pain management practices or the role of religion falls outside the present paper’s scope.

1) In Results, we added a brief linking sentence after the “left to cope alone” quotations to make explicit that these accounts reflect expectations to be strong and to self-manage distress, as described by the women. See lines 429-431.

2) In Implications, we added “as described by participants in this study” after the phrase “cultural needs of Black women” to make the provenance unmistakable. See lines 602-603.

3) In the Introduction (see lines 102-104) and Analysis (see lines 250-254), we state the existing statements that the paper draws on a larger dataset and that other thematic areas are presented elsewhere, so readers understand why provider-specific domains like analgesia practices or religious interface are not expanded here.

Round 2

Reviewer 2 Report

Comments and Suggestions for Authors

Thank you for the responses to my initial review and revisions offered in this manuscript. I reflected upon on what you said and reviewed your manuscript with this in mind. 

Whilst I still think the mention of 'Black women/women' in some places is confusing in the context of what you've presented, in other areas, it does make sense because it relates specifically to the content under discussion. It is undeniable that the study does offer (for example) "insights into the lived experiences of Black women" but where I still find myself at the proverbial crossroads is the extent to which what you've described relates to them as Black women, them as women etc. but I do appreciate that some of the language in your manuscripts has been adjusted to reflect this.

Perhaps what would improve the paper still and relate it a bit more to the experience/needs of Black women is if you offered slightly more clear cut examples. E.g., "Specialized bereavement support that reflects the cultural needs...": what are the cultural needs and what might specialist bereavement support in this context look like? Are you talking about the options you mentioned in the preceding sentence or something else? These, in my opinion, seem to be missed learning moments to offer the unique context of your sample. I return again to your mention of the SBW archetype, could this be reflected on in your implications and recommendations? You mention "integrating culturally responsive services" but what might this mean? 

Broadly speaking, I suppose what I'm seeking is for you to contextualize your suggestions a bit more. They seem to be quite general with some added nuance included following each point but why not target them specifically to your sample? 

e.g., What does training maternity care providers in cultural safety and bias recognition mean? I think it's clear what it might achieve but might be an example of that?

Again, I want to be clear in that I do agree with your implications and recommendations so I'm not disagreeing with any of them.

Author Response

Reviewer comment: Whilst I still think the mention of 'Black women/women' in some places is confusing in the context of what you've presented, in other areas, it does make sense because it relates specifically to the content under discussion. It is undeniable that the study does offer (for example) "insights into the lived experiences of Black women" but where I still find myself at the proverbial crossroads is the extent to which what you've described relates to them as Black women, them as women etc. but I do appreciate that some of the language in your manuscripts has been adjusted to reflect this.

Author response: We appreciate this clarification request. Our analysis centers cross-cutting mechanisms described by a cohort of Black women experiencing IPV and perinatal loss. We do not claim that these mechanisms are unique to Black women. Rather, we situate them within contexts that disproportionately affect Black women in the United States, consistent with established evidence on the social determinants of health. To make this positioning explicit, we added brief clarifying sentences in the Introduction, Results, and Discussion. These changes do not alter the findings; they clarify how to interpret them.

1. Introduction
See lines 104-107

2. Results
See lines 294-296

3. Limitations

See lines 575-581

4. Implications

See lines 590-592

Reviewer Comment: Perhaps what would improve the paper still and relate it a bit more to the experience/needs of Black women is if you offered slightly more clear cut examples. E.g., "Specialized bereavement support that reflects the cultural needs...": what are the cultural needs and what might specialist bereavement support in this context look like? Are you talking about the options you mentioned in the preceding sentence or something else? These, in my opinion, seem to be missed learning moments to offer the unique context of your sample. I return again to your mention of the SBW archetype, could this be reflected on in your implications and recommendations? You mention "integrating culturally responsive services" but what might this mean? 

Broadly speaking, I suppose what I'm seeking is for you to contextualize your suggestions a bit more. They seem to be quite general with some added nuance included following each point but why not target them specifically to your sample? 

e.g., What does training maternity care providers in cultural safety and bias recognition mean? I think it's clear what it might achieve but might be an example of that?

Author Response: Thank you. We agree that specifying practical examples will strengthen the paper. We have revised the Implications and Recommendations section to include concrete, sample-anchored actions that reflect what women described: fear of judgment and care avoidance, being left to cope alone, limited follow-up after loss, and barriers linked to housing, safety, transportation, and support. The added text clarifies what “specialized bereavement support” and “culturally responsive services” entail in this context and shows how training in cultural safety translates into practice. We also explicitly address the SBW expectation by recommending approaches that counter unsupported self-reliance and invite help-seeking.

1. Define “specialized bereavement support” for this sample

See lines 635-643

2. Clarify what “integrating culturally responsive services” means

See lines 590-593, 661-664

3. SBW expectation within recommendations

See lines 658-661

4. Cultural safety and bias-recognition training

See lines 623-628

5. Tie the specificity and scope

See lines 594-597